**Perspective**

# Causes and consequences of Arctic amplification elucidated by coordinated multimodel experiments
James A. Screen [1] ✉, Alexandre Audette [2], Russell Blackport [3], Clara Deser [4], Mark England [1], Nicole Feldl [2], Melissa Gervais [5], Stephanie Hay [1], Paul J. Kushner [6], Yu-Chiao Liang [7], Rym Msadek [8], Regan Mudhar [1], Michael Sigmond [3], Doug Smith [9], Lantao Sun [10] & Hao Yu [1]

Human-induced warming is amplified in the Arctic, but its causes and consequences are not precisely known. Here, we review scientific advances facilitated by the Polar Amplification Model Intercomparison Project. Surface heat flux changes and feedbacks triggered by sea-ice loss are critical to explain the magnitude and seasonality of Arctic amplification. Tropospheric responses to Arctic sea-ice loss that are robust across models and separable from internal variability have been revealed, including local warming and moistening, equatorward shifts of the jet stream and storm track in the North Atlantic, and fewer and milder cold extremes over North America. Whilst generally small compared to simulated internal variability, the response to Arctic sea-ice loss comprises a non-negligible contribution to projected climate change. For example, Arctic sea-ice loss is essential to explain projected North Atlantic jet trends and their uncertainty. Model diversity in the simulated responses has provided pathways to observationally constrain the real-world response.

Polar amplification describes the phenomenon that the polar regions warm at a faster rate than the global average in response to increased greenhouse gases[1–3]. It has long been attributed to the surface albedo feedback: diminishing snow and sea ice reflects less incoming solar radiation, promoting warming and further snow and sea-ice loss. This intuitive explanation, however, is not the full picture for a few fundamental reasons. Firstly, polar amplification occurs in climate models even without any loss of ice or snow, albeit with reduced magnitude[4]. Second, feedback analysis of climate model output consistently highlights the lapse rate feedback, which characterises the sensitivity to the vertical temperature structure, as the single most important driver of polar amplification[2]. Lastly, while sea-ice loss is greatest in late summer, polar amplification is most pronounced in winter[5]. We now understand that polar amplification is a coupled atmosphere-ocean-ice phenomenon that operates across the seasonal cycle[6,7]. It is important to note that the framework through which polar amplification is viewed can lead to different conclusions about the importance of different processes, as the relevant feedbacks are interconnected, but interact in complex ways, with the impact of individual feedbacks potentially enhanced through

synergistic interactions[8–10]. Thus, although the processes and feedbacks leading to polar amplification are reasonably well understood, how they interact, their physical interpretation, their relative contributions, and how they lead to differences in the character of amplification between the hemispheres and seasons are not precisely known[6,7].

Polar amplification may trigger remote climate responses in other parts of the world. For example, a direct consequence of Arctic amplification is a reduction in the near-surface meridional temperature gradient at high latitudes, which implies a weaker jet stream through thermal wind balance[11]. A weaker jet stream may in turn affect the propagation of weather disturbances across the Northern Hemisphere midlatitudes, impacting regional climate[12–14]. However, there is a lack of scientific consensus on the details of any potential influence on midlatitudes[11,13]. Major challenges here are to separate cause from effect, and forced changes from internal variability[15,16].

Climate models are useful tools to probe questions of causality and physical mechanisms, as they allow for controlled experiments to isolate and quantify specific forcings or processes. For example, an atmospheric model can be provided with different sea ice conditions to isolate the atmospheric

[1]Department of Mathematics and Statistics, University of Exeter, Exeter, UK. [2]Department of Earth and Planetary Sciences, University of California Santa Cruz, Santa Cruz, CA, USA. [3]Canadian Centre for Climate Modelling and Analysis, Environment and Climate Change Canada, Victoria, BC, Canada. [4]National Center for Atmospheric Research, Boulder, CO, USA. [5]Department of Meteorology and Atmospheric Science, Penn State University, University Park, PA, USA. [6]Department of Physics, University of Toronto, Toronto, ON, Canada. [7]Department of Atmospheric Sciences, National Taiwan University, Taipei, Taiwan. [8]CECI, CNRS, IRD, CERFACS, Université de Toulouse, Toulouse, France. [9]Met Office Hadley Centre, Met Office, Exeter, UK. [10]Department of Atmospheric Science, Colorado State University, Fort Collins, CO, USA. ✉e-mail: j.screen@exeter.ac.uk

response to sea-ice loss. Myriad sea ice perturbation experiments have been conducted, but often with apparently conflicting results[12,13,17]. For example, models seemingly disagree on the sign of regional winter temperature changes over midlatitudes in response to sea-ice loss[17–19]. Some of the apparent differences between models could relate to sampling uncertainty or be due to varying aspects of the experimental design. However, even absent these factors, the forced response may still be model-dependent. Understanding if and why models differ is important to constrain projections of future climate change[20].

To address these questions, the Polar Amplification Model Intercomparison Project (PAMIP) provided a framework for the scientific community to produce and analyse coordinated model experiments[20]. The PAMIP experiments broadly fall into two categories (Fig. 1): experiments with perturbed sea ice cover (prescribed in atmosphere-only experiments and nudged in coupled experiments) but baseline sea surface temperatures (SST), and experiments with perturbed SST but baseline sea ice. Differencing these experiments allows for the quantification of the simulated responses to sea ice, separately in the Arctic and Antarctic, or SST change, absent other factors (Fig. 1). The 'Tier 1' time-slice simulations have been most widely run and utilised. These are year-long simulations with prescribed SST and sea ice representing preindustrial, present, or future time periods, each with at least 100 ensemble members (i.e., repeated simulations), where ensemble members differ only by very small changes in their initial conditions. Such a large sample size ensures a more robust

quantification of the simulated response to sea ice and SST change in the face of internal variability. This, in turn, allows for a comparison of the forced response across models to quantify model uncertainty and, through the identification of emergent relationships across models and the application of observational constraints, to potentially narrow uncertainties in the real-world response.

The purpose of this Perspective is to synthesise recent advances in understanding the causes and consequences of polar amplification facilitated by the PAMIP. It is not our intention to provide a comprehensive review of polar amplification, which can be found elsewhere[6,7,12,13]. Instead, we draw upon and summarise the latest developments in the field (focussing on work published since the inception of PAMIP in 2019) to identify key advances, remaining questions and pathways to address these, including the need for new coordinated model experiments.

## Causes of Arctic amplification

The PAMIP experiments serve as an ideal testbed to further understand the processes responsible for Arctic amplification. Jenkins et al. (2024) used the PAMIP experiments to show the roles of Arctic sea-ice loss and global SST change in contributing to different local feedbacks and remote processes[21]. Arctic warming in response to sea-ice loss maximises in winter, due to greatly enhanced oceanic heat release. This produces lower tropospheric Arctic warming and triggers positive lapse rate, Planck, and cloud feedbacks, leading to large Arctic amplification (Fig. 2a, b). Despite strong albedo

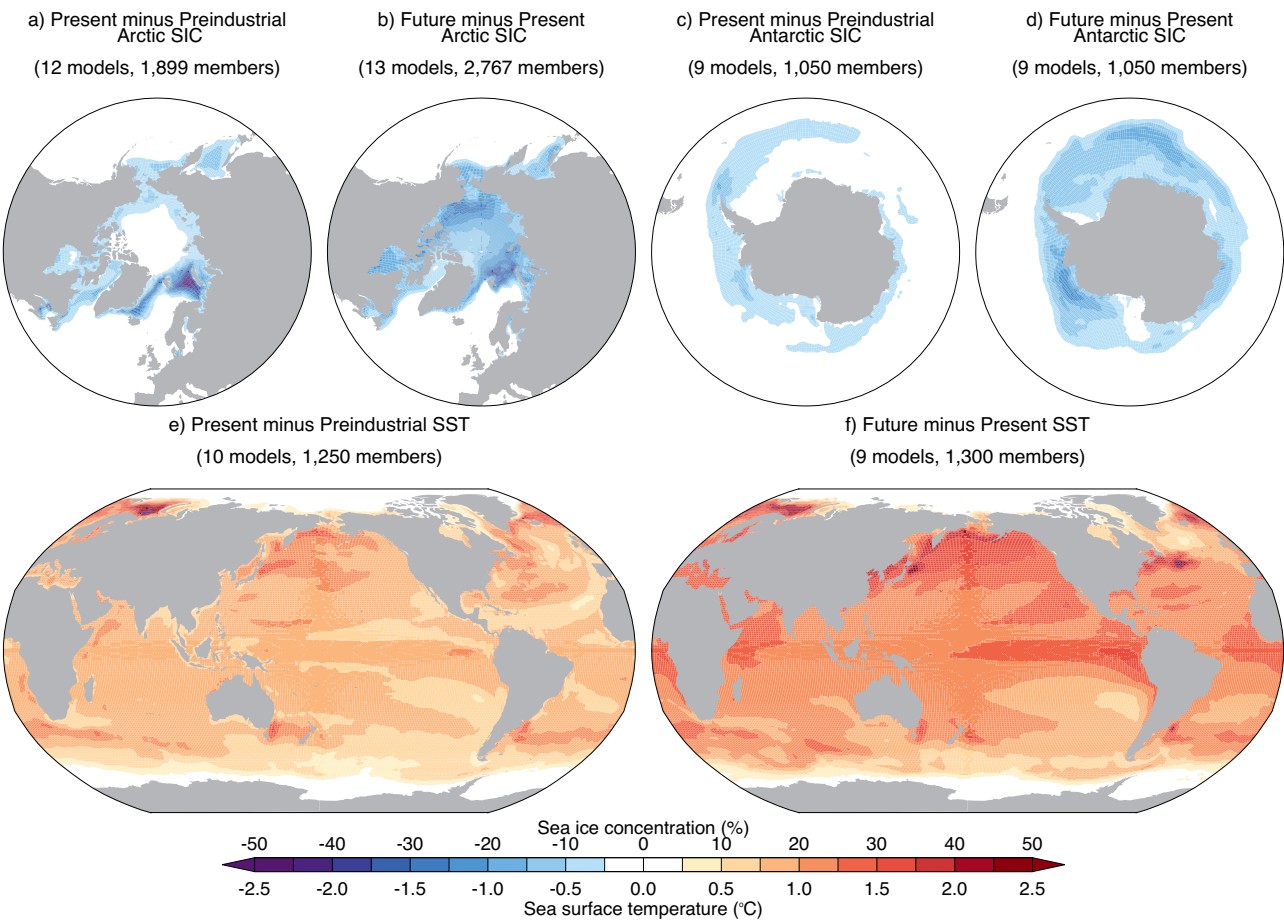

**Fig. 1 | Differences in the specified sea ice and SST between selected PAMIP experiments.** Differences in the prescribed annual-mean sea ice concentration (SIC) and sea surface temperature (SST) between PAMIP experiments to quantify the effects of **a** past and **b** future Arctic sea-ice loss; **c** past and **d** future Antarctic sea-ice loss; and **e** past and **f** future SST change. The number of models providing each experiment combination and the total number of members, for each year-long time-slice experiment, across all models (publicly available on the Earth System Grid Federation) are provided. The present-day SIC and SST are representative of the period 1979–2008. The pre-industrial and future periods correspond to when the global mean temperature was 0.6 °C cooler and 1.4 °C warmer than present-day (2 °C warmer than preindustrial), respectively.

**Fig. 2 | Contributions to seasonal Arctic warming from various processes and feedbacks.** Arctic warming contributions (in K) in the warm half-year (April-September) *vs.* cold half-year (October-March) from the surface albedo (α, red), water vapor (q; blue), Planck (PL'; black), lapse rate (LR; green) and cloud (C; cyan) feedbacks, and changes in oceanic heat release (-ΔOHU; orange) and atmospheric energy convergence (Δ(-∇·F$_A$); maroon) in response to **a** past and **b** future Arctic sea-ice loss, and **c** past and **d** future SST warming. Adapted from Jenkins et al. (2024)[21].

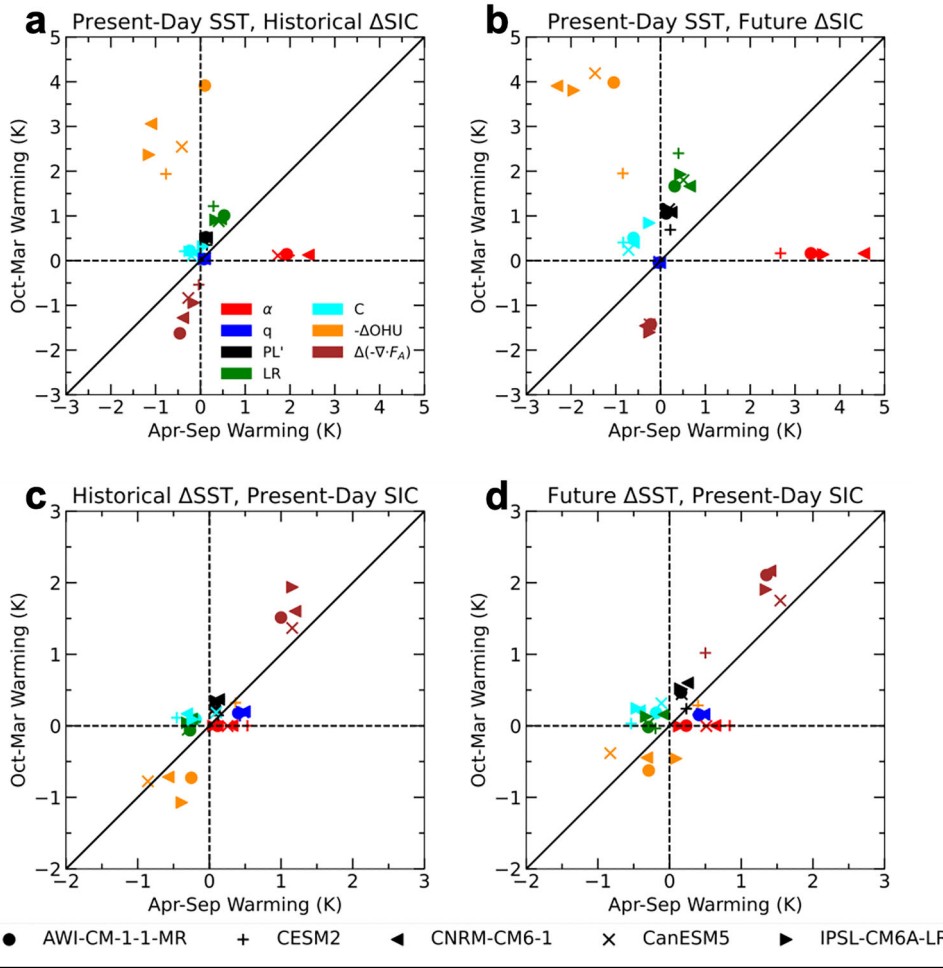

feedback in summer, atmospheric warming is muted in that season. In contrast, in response to global SST warming, absent sea-ice loss, enhanced atmospheric energy convergence into the Arctic is the dominant contributor to Arctic warming, although Arctic amplification is relatively small compared to that in response to sea-ice loss (Fig. 2c, d). Here, the lapse rate feedback is of diminished importance, consistent with SST-induced warming being larger aloft than at the surface. In all cases, the water vapor feedback contributes to Arctic warming but opposes amplification due to larger tropical than Arctic moistening under SST-induced warming with fixed Arctic sea-ice. These results reinforce that changes in surface fluxes and feedbacks triggered by sea-ice loss are critical to explain the magnitude and seasonality of Arctic amplification, while increased poleward energy transport produces weaker amplification in the absence of sea-ice-related feedbacks. We emphasise that Fig. 2 presents a diagnostic decomposition of the contributions to Arctic amplification and in reality, the effects of sea-ice loss and SST warming act in combination and interact.

While the magnitude and seasonality of polar amplification are controlled by local processes and feedbacks (Fig. 2), disproportionately large Arctic warming is a fundamental response of a moist atmosphere to an increase in greenhouse gases[22,23]. A preferential increase in tropical humidity, which occurs due to the Clausius-Clapeyron relation, provokes an increase in poleward atmospheric latent energy transport. This process explains polar amplification in the absence of sea ice and related polar feedbacks[4,24], and the greater amplification when local and remote drivers combine[22]. PAMIP has expanded understanding of these poleward energy transport changes by linking them to the processes of sea-ice loss and SST changes. Audette et al. (2021) show that Arctic sea-ice loss reduces northward eddy-driven energy transport into the Arctic in all the PAMIP models, owing to the reduced near-surface temperature gradient, which is balanced

by a similar magnitude increase of latent energy transport due to SST warming[25]. This opposition in total energy transport mainly arises in the annual mean as the influence of Arctic sea-ice loss is greatest in winter, while that of SST warming is greatest in summer[26]. Hence, there is growing appreciation that different aspects of the poleward energy transport exhibit different efficacies, and that the polar-cap-averaged energetic perspective risks obscuring the role of remote processes in driving polar amplification.

Precipitation changes are also amplified in the Arctic relative to lower latitudes[27]. The PAMIP experiments have shown that sea-ice loss and SST warming both increase Arctic precipitation, but through predominantly different mechanisms. SST warming increases precipitation in the Arctic[28], consistent with the increase in moisture convergence in the same experiments[25,26], while sea-ice loss also increases precipitation through increased evaporation[21,28,29]. Since SST warming alone leads to larger wetting at lower than higher latitudes, it is sea-ice loss that is critical for amplified wetting in the Arctic[29].

## Robust remote responses to sea-ice loss

Prior to the PAMIP, the precise nature of the remote circulation response to sea ice loss, including the effect on the jetstream, storm tracks, and dominant modes of variability (e.g., North Atlantic Oscillation, Northern Annular Mode), as well as the mechanisms involved, was elusive[13,20]. This is due to a variety of factors, including, but not limited to, inter-model differences, differences in experimental design, and sampling errors due to large internal variability. The PAMIP has facilitated like-for-like comparison across models (owing to identical boundary conditions) and better separation of the forced response from internal variability (by requiring a minimum of 100 ensemble members), although even larger ensembles appear to be necessary to confidently capture the stratospheric response[30–32] and changes

in regional weather extremes[33–35], as discussed later. One important advance from the PAMIP has been to identify robust large-scale troposphere circulation responses to future Arctic[36] and Antarctic sea-ice loss[37]. More specifically, the PAMIP experiments impose changes in sea ice concentration (hence, areal coverage; Fig. 1) throughout the year and so, include shifts in the seasonal cycle, but not changes in sea ice thickness.

Smith et al. (2022) examined the PAMIP 'present-day' baseline and future Arctic sea ice experiments from 16 atmospheric models and showed a robust equatorward shift of the tropospheric westerly jet in response to future Arctic sea-ice loss, albeit of varying magnitudes between models[36]. A similar equatorward shift of the wintertime jet was found in the southern hemisphere in response to future Antarctic sea-ice loss, also robust in sign but of varying magnitude across models[37]. The mechanisms that lead to an equatorward jet shift are shown schematically in Fig. 3. Briefly, Arctic warming induced by sea-ice loss reduces the high-latitude meridional temperature gradient (step 1 in Fig. 3), reducing wind speed on the poleward flank of the jet (following the thermal wind relationship). The weakened temperature gradient also reduces baroclinic eddy activity, weakening the storm track and, in turn, reducing the upward wave activity flux from the surface (step 2; consistent with reduced eddy-driven poleward heat transport). An anomalous meridional circulation develops with ascent around 40-50 °N, poleward flow in the mid to upper troposphere, descent around 65–75 °N, and equatorward flow near the surface (step 3). Adiabatic cooling of the ascending branch acts to enhance the latitudinal temperature gradient on the equatorward side of the jet, which strengthens the wind to the south of the jet core (step 4). Taken together, the weakened westerlies to the north and strengthened westerlies to the south imply an equatorward shift of the jet and storm activity (step 5). The shifted storm activity leads to wave flux anomalies that reinforce the jet shift through positive eddy feedback (step 6). The magnitude of the response across models appears to depend on the strength of this atmospheric eddy feedback, in both hemispheres[36,37]. This zonal-mean perspective also applies for the Atlantic basin. In the Pacific,

while the anomalies are broadly similar, the climatological jet is located farther south, so the westerly wind anomalies act to strengthen the Pacific jet rather than shift it equatorward[38].

Regionally, the winter North Atlantic jet shifts equatorward in response to Arctic sea-ice loss in most models, with a few models showing a negligible response[30,35,40]. However, models disagree on the sign of simulated changes in the speed and tilt of the North Atlantic jet, with most models showing negligible change[35,39,40]. In contrast, the North Pacific jet is simulated to strengthen in response to future Arctic sea-ice loss, on average across models[38,41]. The contrasting jet speed responses in the North Atlantic and North Pacific can be understood by the more equatorward climatological position of the North Pacific jet, such that the strengthened westerly wind in response to sea-ice loss occurs in the core of the jet in the Pacific rather than on the equatorward flank, as simulated in the Atlantic[38,42].

Accompanying the jet shift is a weakening and equatorward shift of the North Atlantic storm track[34,38,43,44]. An eastward shift or extension of the North Pacific storm track is consistent with a strengthened and extended jet[38,41]. Across the whole Northern Hemisphere, in response to future Arctic sea-ice loss, there are fewer individual storms simulated, those storms are weaker, they propagate more slowly, and have longer lifetimes[43], which can be understood as an energetic response to change in surface albedo and weakened poleward atmospheric energy transport[45]. However, over North America, weather systems do not appear to stagnate, although they tend to be weaker[33]. Other weather system responses to Arctic sea-ice loss identified using the PAMIP experiments include an increase in winter Scandinavian blocking[35,46], a reduction in cut-off-lows over Southern Europe[47], enhanced tropical cyclone genesis over the eastern North Pacific[48], and shifts in the predominant locations of atmospheric rivers[49], albeit with varying degrees of confidence (Fig. 4).

Drying over Northwest Europe and wetting over central Europe are found in response to future Arctic sea-ice loss, closely linked to storm track changes[34,35,39,50]. Warmer and fewer cold air outbreaks over central and

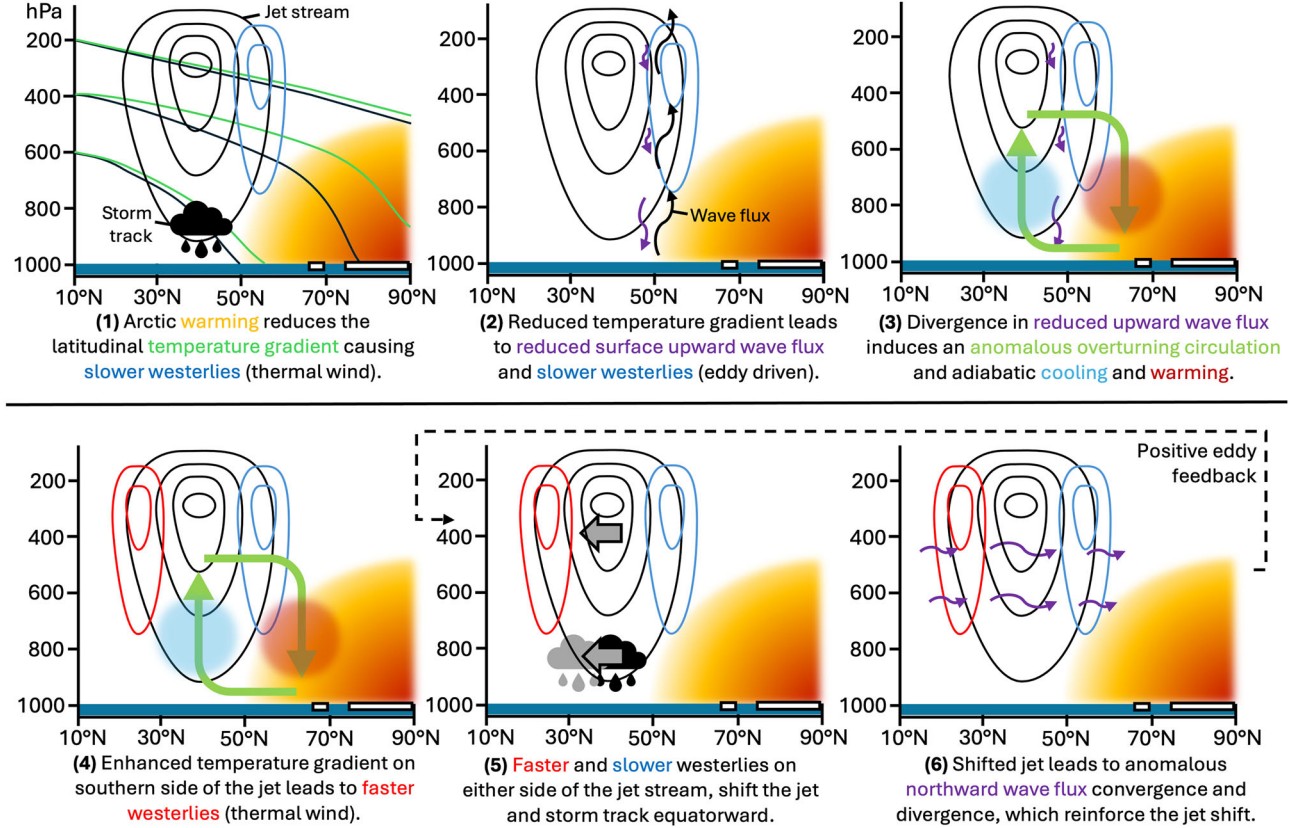

**Fig. 3 | Schematic representation of the mechanisms of the jet shift in response to Arctic sea-ice loss.** Features depicted in black represent the climatological state whereas those shown in colour represent the response to sea-ice loss. See Smith et al. (2022)[36] for further details.

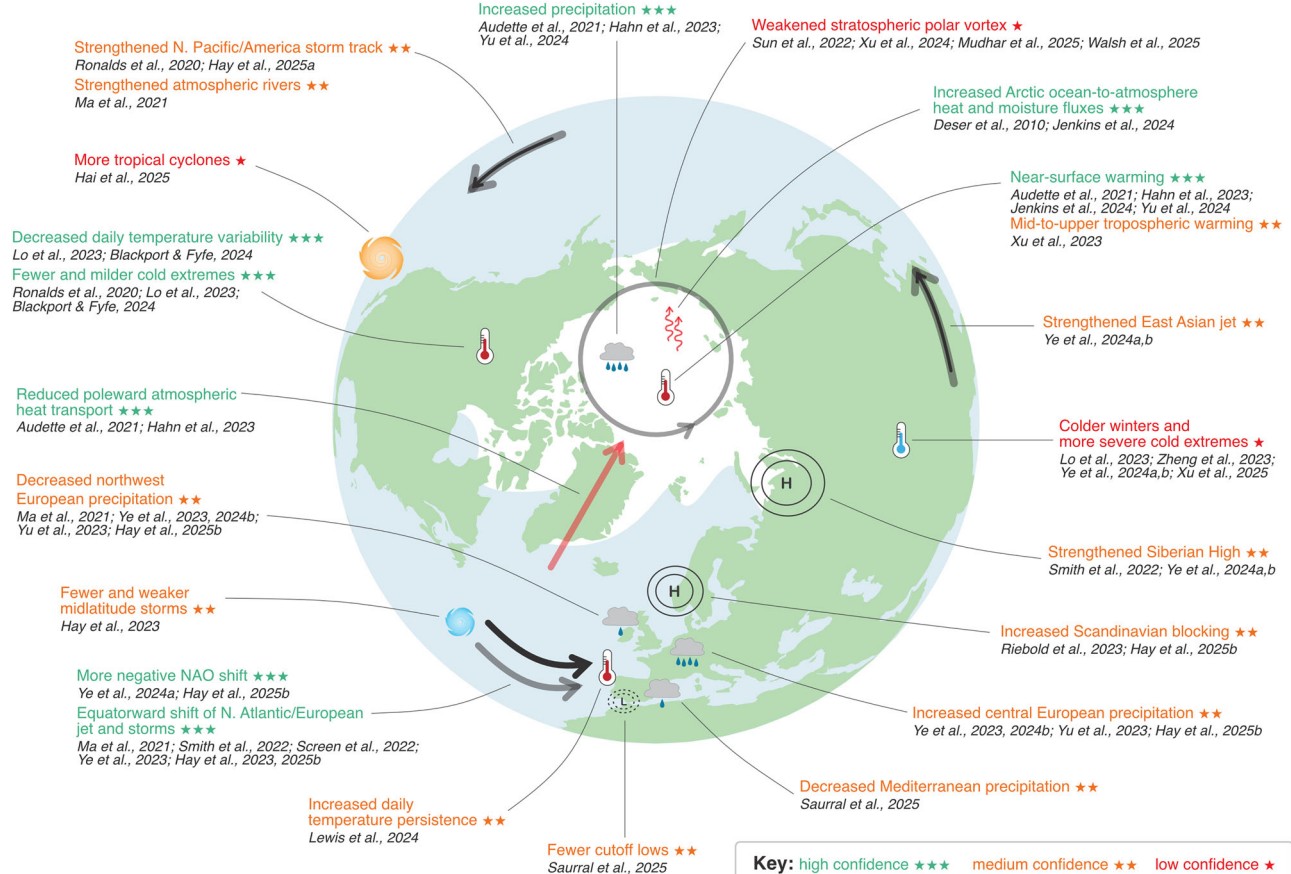

**Fig. 4 | Regional effects of Arctic sea-ice loss.** Visual summary of the predominantly wintertime effects of Arctic sea-ice loss across the Northern Hemisphere, as simulated in the PAMIP experiments. The assessed confidence is based on expert judgement, considering the level of consistency between studies/models/experiments, knowledge of the mechanisms involved, and evidence that the simulated response is representative of that in the real world. The locations of features are approximate. The supporting references are examples and are not intended to provide a comprehensive list of all studies that have reported on each effect.

western North America are connected to the strengthened and extended North Pacific jet[41]. Reduced meridional winds are linked to increased surface temperature persistence[51]. In addition to these dynamically driven responses, advection across the weakened meridional temperature gradient causes a reduction in day-to-day temperature variability and more rapid warming of cold extremes than of the average temperature[52], consistent with earlier findings[53]. Cold days, which typically occur under northerly flow, warm faster than the seasonal mean (and hot days), as northerly flow advects Arctic air masses that are significantly warmed by sea-ice loss to lower latitudes[52]. Lo et al. (2023) found the 1-in-20-year extreme cold events over northern mid-to-high latitudes warmed by up to 2.5 °C for eastern Canada and the northeast United States[54].

The varied and widespread responses to Arctic sea-ice loss, summarised visually in Fig. 4, have been predominantly identified in winter, although some responses have been reported in summer[48,55–58]. The larger magnitude responses in winter than summer[9,28,43], arise due to the seasonality of the Arctic surface energy budget response to sea-ice loss[9,59]. Anomalous energy transfer from the ocean to the atmosphere maximises in the winter, as does the Arctic warming response, which is the trigger for the subsequent effects on the atmospheric circulation and midlatitude weather and climate.

## Reducing and exploiting uncertainty

Atmospheric internal variability has emerged as a common challenge in understanding the character of the forced response to sea-ice loss, especially for applications involving regional climate and extremes[33–35,41,49,60] and the stratospheric polar vortex[30–32,61,62]. The PAMIP protocol recommended a minimum ensemble size of 100 members. However, Peings et al. (2021)

found substantial inconsistencies in the atmospheric circulation response to Arctic sea-ice loss among three separate 100-member ensembles, indicating that ensembles of this size still contain a substantial imprint of internal variability[30]. Similarly, Sun et al. (2022) showed that the stratospheric polar vortex response in PAMIP simulations is small compared to its internal variability[31]. Even the sign of the stratospheric circulation response to sea-ice loss is uncertain[36], for reasons that remain unclear but are likely to include low signal-to-noise and state dependence[32,62,63], and in turn influences the magnitude of the tropospheric circulation response[31,32,62,63]. Ye et al. (2024) is unique in conducting very large ensembles (~2000 members) with two models following the PAMIP protocol and found that several hundred members (>400) are needed to robustly estimate the seasonal-mean large-scale circulation response, and a thousand or more members for regional climate extremes[34]. This could imply the real-world response to sea-ice loss is weak[15,36,37,63]. However, the weak simulated responses in the PAMIP experiments could be a symptom of a broader signal-to-noise problem in models[64] and thus, the real-world response may be larger than models suggest.

Internal variability appears insufficient to fully explain the model spread in the magnitude of the tropospheric circulation response[36,37,63], suggesting this response is model dependent. One factor that may lead to model dependence is the strength of atmospheric eddy feedback, which describes the reinforcing interaction of transient eddies on the time-mean flow. Models exhibiting stronger eddy feedback simulate a stronger tropospheric circulation response to Arctic and Antarctic sea-ice loss[36,37]. Understanding the causes of model spread is valuable not only for interpreting model responses but also for reducing uncertainty in projections through an emergent constraint—an inter-model relationship linking an

observable aspect of the climate to the response to sea-ice loss. One example is the relationship between the simulated eddy feedback and the jet response to sea-ice loss across the PAMIP models[36,37]. The observed estimate of eddy feedback is larger than any of the modelled values, which suggests that models systematically underestimate the atmospheric circulation response to Arctic sea-ice loss. While there are suggestions that simulated eddy feedback is too weak owing to coarse model resolution[65], it remains unclear if the response to sea-ice loss is dependent on model resolution[61]. Constraining by the observed eddy feedback increases the equatorward jet shift in response to Arctic sea-ice loss by 40% compared to the multimodel mean, becoming of larger magnitude than the simulated poleward shift in response to SST warming[37].

Additionally, differences in models' unperturbed climate - often referred to as the basic state - can also influence the response to sea-ice loss[62,63,66]. The basic state of the winds in the so-called "neck region", located in the upper troposphere-lower stratosphere between the latitudes of the subtropical jet and polar jet, determines whether the stratospheric pathway is active, and thereby affects the magnitude of the tropospheric circulation response to Arctic sea-ice loss[62,63,67]. Sigmond and Sun (2025) propose that the jet latitude response to future sea-ice loss can be constrained by the observed neck region winds, more than halving the model uncertainty[63]. The neck wind constraint yields an estimated real-world jet shift that closely matches the unconstrained multi-model mean response, implying that models do not systematically underestimate the jet shift[63], unlike the constraint based on observed eddy feedback[36,37]. These contrasting findings underscore the need to test the robustness of such emergent constraints using independent ensembles. Stratosphere-troposphere coupling also plays a central role in mid-tropospheric Arctic warming in response to sea-ice loss[68], and this deep warming can further enhance the tropospheric circulation response[69].

Slow modes of natural climate variability can alter the 'basic state' and thereby modulate the response to sea-ice loss. Labe et al. (2019) found that the atmospheric circulation response to Arctic sea-ice loss was stronger during the easterly phase of the Quasi-Biennial Oscillation (QBO) than its westerly phase[70]. This led one group, the UK Met Office, to perform separate PAMIP ensembles for different QBO phases. The stratospheric polar vortex weakened in response to Arctic sea-ice loss during easterly QBO but not during westerly QBO[32,71,72]. The different QBO states across the PAMIP models may contribute to the model differences in the responses of the stratospheric polar vortex[32]. There is also evidence that the phase of the El Niño-Southern Oscillation (ENSO), Pacific (Inter-) Decadal Variability (PDV), and Atlantic Multidecadal Variability (AMV) all influence the atmospheric response to Arctic sea-ice loss[30,72–76]. These modes of variability rely on coupling between the ocean and atmosphere. A recent study by Cvijanovic et al. (2025) suggests that the North Pacific response to Arctic sea-ice loss is strongly influenced by ocean–atmosphere coupling in the tropics[77]. Models that prescribe tropical SSTs — thereby lacking ocean-atmosphere coupling - tend to simulate a strengthened Aleutian Low over the North Pacific in response to sea-ice loss[36], as do many coupled model experiments[17], but in the study of Cvijanovic et al. (2025), the inclusion of coupling resulted in a weakened Aleutian Low[77]. We will return to the potential importance of ocean coupling and the limitations of prescribing climatological SST (i.e., lacking internal ocean variability) later.

## Interpreting climate projections

Confidence in climate projections is increased when the underpinning physical mechanisms are well understood. Since sea-ice loss and SST warming may affect the atmospheric circulation in different ways and through different mechanisms, it is of value, therefore, to consider the responses to these factors separately, which the PAMIP experiments have facilitated. The effect of SST warming dominates over most of the globe, outside the Arctic[9,28] (Fig. 5). However, the effect of sea-ice loss is as important, if not more, than SST warming for the response of the winter North Atlantic circulation[28,35,38] (Fig. 5) and East Asian summer monsoon[58].

Separation of the responses to SST warming and sea-ice loss has been particularly useful to uncover the mechanisms driving the extratropical circulation changes. Using the PAMIP experiments, Yu et al. (2024) demonstrated a significant regional weakening of the westerlies over the high-latitude North Atlantic and strengthening over the midlatitude North Atlantic in response to Arctic sea-ice loss[28], consistent with the zonal-mean response[36], while ocean warming led to a broadly opposite response[28] (Fig. 5). This regional 'tug-of-war' in the Atlantic is also seen for storm track activity[38,50], but not in the Pacific[28,38,41]. Hay et al (2025) showed that the more equatorward average location of the Pacific jet and the geography of the North Pacific results in circulation responses to sea-ice loss and ocean warming that reinforce rather than oppose each other[38], in contrast to the zonally-averaged response[36].

The role of Arctic sea-ice loss on cold winter extremes, in the context of a warming climate, has been the subject of debate for the past few decades[12–15]. The PAMIP has facilitated a clearer picture of the relative roles of SST warming and Arctic sea-ice loss on mid-latitude extreme cold events. While Arctic sea-ice loss leads to more frequent or severe cold extremes in some simulations in limited midlatitude regions[19,54], these changes are small and overwhelmed by strong warming of extreme cold events over the midlatitudes in response to ocean warming[46,54]. However, sea-ice loss plays a more dominant role than SST warming in the reduction of temperature variability over North America[52].

It is worth noting that the PAMIP experiments are based upon one scenario of sea-ice loss and SST warming, but model projections suggest a range of outcomes. In the PAMIP experiments, comparing the annual-mean future and present-day states, there is a loss of 3.0 million km² of sea ice cover for a global SST warming of 1.4 C, equating to approximately 2.1 million km² of sea-ice loss per degree Celsius of global SST warming. Model projections show sea-ice sensitivities of less than 1 to more than 4 million km² per degree of global SST warming[35]. Under the assumptions of linear additivity and scalability[29], the PAMIP-derived responses to sea-ice loss and SST warming can be scaled and summed to illustrate different storylines. Storylines are a concept to help communicate uncertainty in climate change[78] and describe distinct but possible, physically self-consistent pathways. Here, the storylines describe plausible combinations of future global warming and sea-ice loss (Fig. 6), which are closely related to model uncertainty in the magnitude of Arctic amplification, expressed in terms of the magnitude of sea-ice loss per degree of global warming. The high sensitivity case describes a future with greater sea-ice loss relative to global warming, and the low sensitivity case describes a world with less sea-ice loss relative to global warming. Arctic warming, Arctic amplification (2.7 and 4.2 in low and high sensitivity cases, respectively,) and Arctic wetting (0.12 and 0.18 mm/day, respectively) increase with greater sea-ice sensitivity to global warming (Fig. 6). The magnitude and pattern of the circulation response at lower latitudes are also sensitive to the storyline. For example, the North Atlantic jet shifts marginally poleward (0.1 ° latitude) in the low sea-ice sensitivity storyline and equatorward (-0.5 °) in the high sea-ice sensitivity storyline. Given that models appear to underestimate the observed sea-ice sensitivity[79], the high sea-ice sensitivity storyline may be the most plausible.

## Future directions
To close, we highlight some future research opportunities.

## Hemispheric asymmetry
First, polar amplification exhibits notable hemispheric asymmetry, with the Arctic experiencing markedly stronger warming than the Antarctic[2]. Several mechanisms underpinning this disparity have been proposed, including a buffering effect by the presence of a deep and cold Southern Ocean, which limits surface warming through effective oceanic heat uptake[80], and high topography in the Antarctic that inhibits temperature longwave feedbacks[1]. At a longer timescale, oceanic heat transport contributes to this hemispheric asymmetry, as the AMOC facilitates poleward heat transfer and local feedbacks in the Northern Hemisphere[81], whereas analogous mechanisms

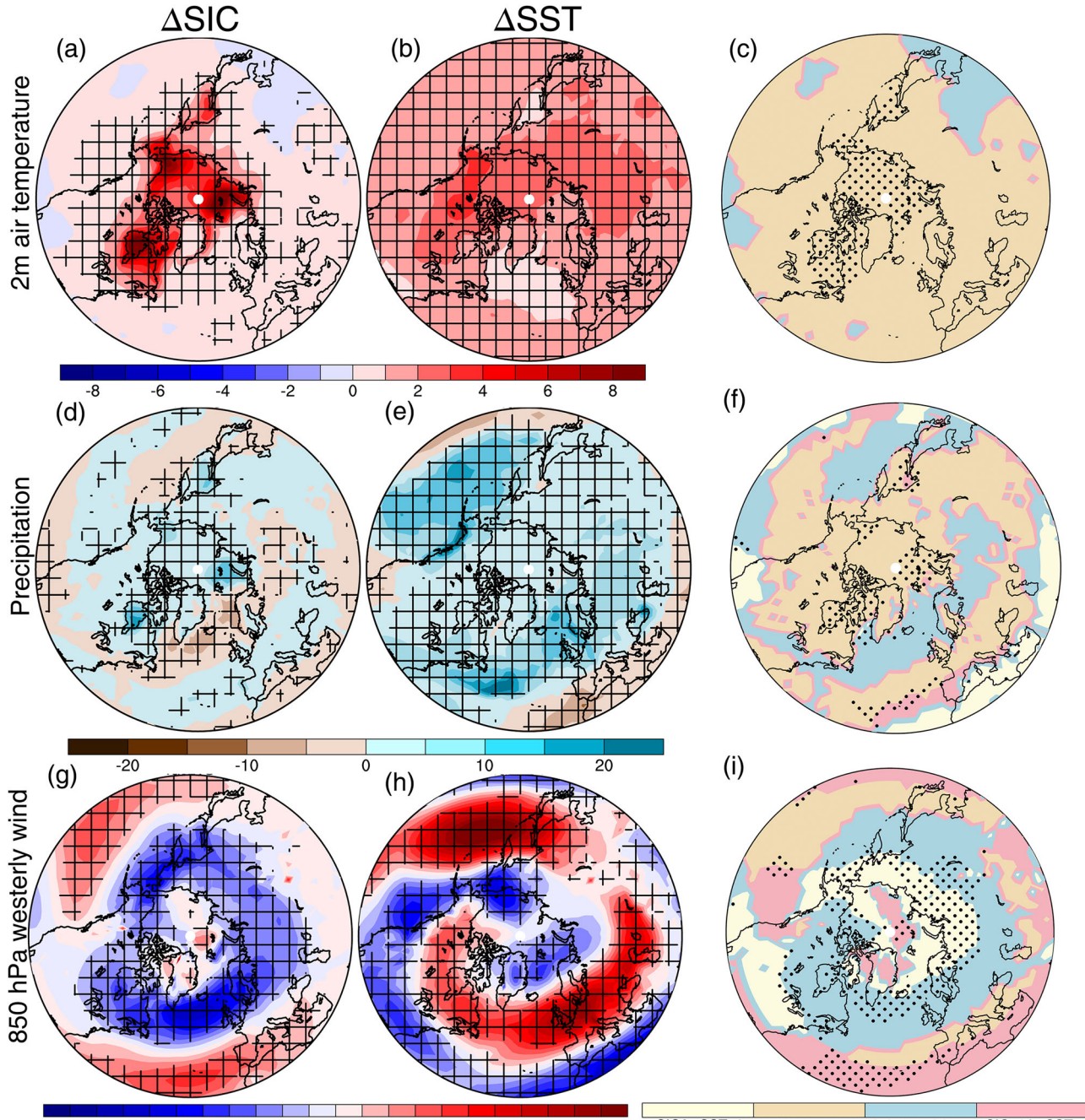

**Fig. 5 | Distinct responses to sea-ice loss and SST warming.** Multimodel-mean winter near-surface air temperature response to **a** future Arctic sea-ice loss and **b** future SST warming, and **c** their relative signs and magnitudes. Hatching denotes statistical significance at the 90% confidence level. In **c**, yellow and orange colors denote the responses reinforce each other, whereas blue and red colors denote the responses oppose each other, and stippling shows where the response to sea-ice loss is of greater magnitude than that to SST warming. **d–f** As **a–c**, but for precipitation. **g–i** As **a–c**, but for 850 hPa westerly wind. Adapted from Yu et al. (2024)[28].

are weaker in the Southern Hemisphere. Beyond temperature change, Arctic precipitation amplification - driven by increased moisture availability and enhanced poleward moisture transport that remains energetically constrained[27,82] - enhances Arctic hydrological sensitivity and serves as another manifestation of polar amplified climate change. In sum, the interplay between sea-ice feedbacks, ocean dynamics, and atmospheric processes leads to pronounced and hemispherically asymmetric polar climate responses. We urge more process-based analysis to improve our understanding of asymmetries in the two poles and their implications for the global climate. This need is particularly urgent given the unprecedented Antarctic sea-ice loss since 2016, which may signal a transition to a new

Antarctic climate regime[83–85], for which the associated atmospheric response, both locally and globally, remains relatively unknown[86,87]. Advancing our understanding of these processes is critical for anticipating future climate impacts.

## Experimental protocols

Within the framework of the PAMIP, a variety of experimental strategies have been employed to advance the understanding of polar climate processes and their remote influences, including prescribing ocean surface boundary conditions (in atmosphere-only experiments) and nudging or albedo reduction techniques (in coupled atmosphere-ocean-ice

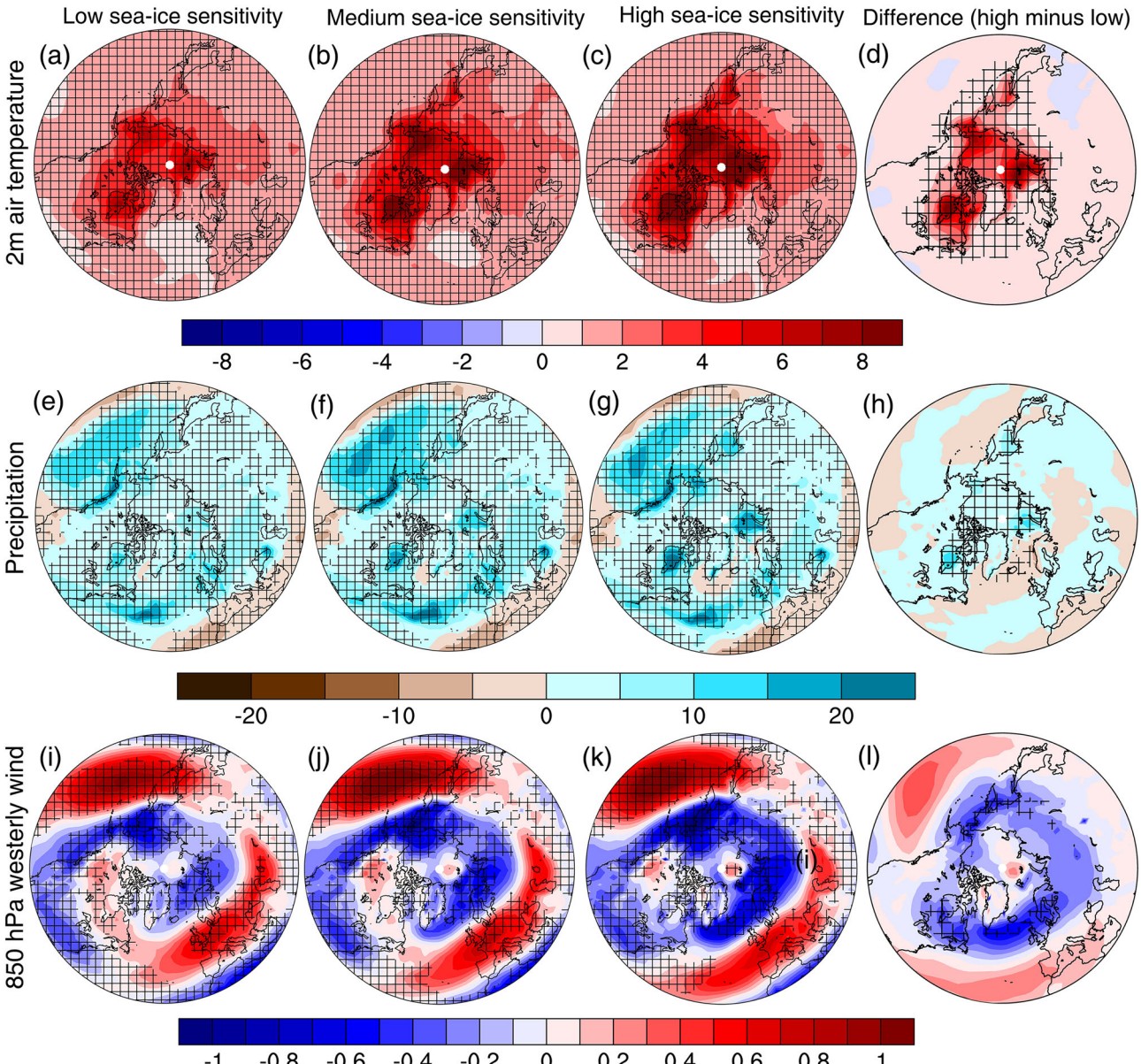

**Fig. 6 | Storylines of projected climate change. a–c** Summed multimodel winter near-surface air temperature responses to future Arctic sea-ice loss and SST warming, for three different storylines: low sea-ice sensitivity (1.5 million square kilometers of sea-ice loss per degree Celsius of global-mean SST warming), medium sea-ice sensitivity (2.7 million km²/°C) and high sea-ice sensitivity (3.8 million km²/°C). The low and high sensitivity storylines are calculated by linearly scaling the PAMIP responses prior to their summation; the medium sea-ice sensitivity storyline uses the unscaled PAMIP responses. **d** The difference between the high and low sensitivity storylines. **e–h** As a-d, but for precipitation. **i–l** As **a–d**, but for 850 hPa westerly wind. Hatching denotes statistical significance at the 90% confidence level. Numbers in the lower left corner denote Arctic amplification in **a–c**, Arctic-mean precipitation change in **e–g**, and the North Atlantic jet latitude change in **i–k**. Adapted from Yu et al. (2024)[28] and Hay et al. (2025)[35].

experiments). Ocean coupling appears to modulate the response to sea-ice loss, and aspects of the response may be underestimated by prescribing sea ice and thereby inhibiting ocean-ice-atmosphere interaction[42,44,86,88–91]. The PAMIP coupled experiments simulate enhanced weakening of the winter storm tracks in response to sea-ice loss, compared to the uncoupled experiments[44], and deeper Arctic warming and enhanced strengthening of the Siberian High[91]. However, concerns have been raised that the methods used to constrain sea ice in coupled modes might lead to spurious responses[92–95]. New methods for addressing these issues are necessary for developing protocols for the next phase of the PAMIP. For example, the climate impacts of spurious heating associated with current nudging techniques can be quantified, and hence removed, through additional model simulations[93] or through post-processing via pattern scaling[95]. Alternatively, regional $CO_2$ forcing may offer a pathway for isolating Arctic change with fewer unintended consequences and is technically more straightforward to implement across all climate models than nudging[96,97]. Other methods, such as those involving modifications to sea ice and snow-over-sea-ice parameters[77,85,98] or atmosphere-ocean-ice coupling[99], have been proposed and may reduce any spurious responses. The most utilised experiments, the year-long atmosphere-only simulations, not only omit ocean coupling, but also lack ocean internal variability (as climatological boundary conditions are prescribed) and processes acting on interannual timescales. Transient experiments, although more costly to run, allow for better sampling of internal variability and response timescales. Additionally, considering different initial states in the presence of sea ice forcings, such as sampling opposite phases of the QBO, ENSO, AMV, or PDV, would further enhance

our understanding of the state dependency (i.e., nonlinearity) of responses to sea-ice loss[70–76]. Greater coordination of atmospheric and ocean initial conditions across modelling protocols may help to explain apparent model discrepancies that remain hard to reconcile, for example, the divergent stratospheric polar vortex responses across models[32].

## New models and experiments

High-resolution models, either globally or with regional grid refinement, better capture fine-scale processes and therefore further improve the representation of sea-ice-atmosphere interactions and feedbacks[29,100] and simulate stronger circulation responses to SST anomalies[101]. However, whether the atmospheric response to sea-ice loss is sensitive to model resolution remains unclear[29,32,61] and warrants further investigation. Additionally, saving more diagnostic variables at high temporal frequency (daily or 6-hourly), would provide opportunities to examine processes at the scale of weather events. New experiments with different prescribed boundary conditions (e.g., larger global warming and sea-ice loss to increase the signal-to-noise ratio, and different storylines of sea ice and SST change) may also yield important insights. Repeating the existing experiments with newer models, including at higher resolution, would be valuable to test the robustness of emergent constraints with independent ensembles. Idealized simulations also continue to serve as a useful tool for conceptual understanding, providing controlled environments to disentangle nonlinear interactions and benchmark model behaviour across scales[1,4,21,22,24,45,51,67,94,102]. Such modelling experiments offer opportunities to target remaining questions regarding the causes of polar amplification, such as the interpretation of the lapse rate feedback in polar regions. More recently, Artificial Intelligence and machine learning-based climate models[103] offer new avenues to simulate complex climate dynamics with enhanced computational efficiency, although their capacity to fully capture sea-ice thermodynamics and large-scale coupling remains under assessment.

## Enhanced collaboration

We encourage the use of PAMIP simulations for novel and varied applications. Examples include interpreting the implications of incomplete Arctic sea-ice recovery under $CO_2$ removal[104], exploring the role of sea-ice-related feedbacks in sustaining Barents Sea ice loss[105], identifying the surface signature of stratospheric variability[106], and quantifying the effects of projected sea-ice loss on cold-related mortality[107], Greenland ice sheet mass balance[57], Yangtze River Basin heatwaves[108] and eastern Siberian wildfires[109], amongst others. Expanding collaborations with other research communities will be critical for the next phase of PAMIP. Coordination between future PAMIP and other model intercomparison projects could enhance process-based understanding. Joint efforts with the climate feedback, atmosphere-ocean-ice dynamics, and polar process communities could further enrich the interpretation of results. Side-by-side analysis of modelling experiments within dynamical and radiative feedback frameworks will provide deeper insights into the processes and mechanisms by which a climate response to sea ice loss is provoked. Closer ties with the polar observational community would support robust model–observation comparisons, which likely need better experimental designs and additional high-frequency model outputs.

In summary, the PAMIP has facilitated a plethora of studies (not all are cited here), but arguably the biggest collective advance has been to reveal robust tropospheric responses to Arctic sea-ice loss, that whilst generally small compared to simulated internal variability, comprise a non-negligible, and for some regions and variables, large (relative to that of SST warming), contribution to projected climate change. Further, it has uncovered and explained responses that are dependent on model physics and/or background states, which provide pathways to constrain the real-world response. Looking forward, there are opportunities, through novel analyses of existing experiments, new simulations and model versions, and enhanced collaborations, to gain further insights into the causes and consequences of polar amplification.

## Data availability

All PAMIP data analysed here is freely available at the Earth System Grid Federation (https://esgf.github.io).

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

## Acknowledgements
Matt Jenkins is thanked for providing Fig. 2. We thank the modelling groups that have contributed to the PAMIP, and the Earth System Grid Federation (https://esgf.github.io) for providing the infrastructure for data storage and sharing. This work used JASMIN, the UK's collaborative data analysis environment (https://www.jasmin.ac.uk). J.A.S. and S.H. were supported by the NERC ArctiCONNECT project (NE/V005855/1). C.D. was supported by the National Center for Atmospheric Research (NCAR), which is sponsored by the National Science Foundation under Cooperative Agreement 1852977. M.E. was supported by the Royal Commission for the Exhibition of 1851 research fellowship. N.F. was supported by NSF Grant AGS-1753034. P.K. acknowledges support from the grant NSF PHY-2309135 to the Kavli Institute for Theoretical Physics (KITP). Y.-C.L. was supported by the National Science and Technology Council (113-2628-M-002-018 and 113-2116-M-008-024). R.M. is funded by a NERC GW4+ Doctoral Training Partnership studentship (NE/S007504/1). D.S. was supported by the Met Office Hadley Centre Climate Programme funded by DSIT. L.S. was supported by NSF Grant AGS-2300038. H.Y. was supported by the Chinese Scholarship Council. We thank three anonymous reviewers for their constructive feedback.

## Author contributions
James A. Screen: conception, analysis, preparation of figures, writing, review and editing. Alexandre Audette: writing, review and editing. Russell Blackport: writing, reviewing and editing. Clara Deser: writing, review and editing. Mark England: writing, review and editing. Nicole Feldl: writing, review and editing. Melissa Gervais: preparation of figures, writing, review and editing. Stephanie Hay: analysis, preparation of figures, writing, review and editing. Paul J. Kushner: writing, review and editing. Yu-Chiao Liang: writing, review and editing. Rym Msadek: writing, review and editing. Regan Mudhar: preparation of figures, writing, review and editing. Michael Sigmond: writing, review and editing. Doug Smith: preparation of figures, writing, review and editing. Lantao Sun: writing, review and editing. Hao Yu: analysis, preparation of figures, writing, review and editing.

## Competing interests
The authors declare no competing interests.
