## [Transparent Peer Review file · Communications Earth & Environment]

Causes and consequences of Arctic amplification elucidated by coordinated multimodel experiments

Corresponding Author: Professor James Screen

Version 0:

Decision Letter:

Dear Professor Screen,

Your manuscript titled "Causes and consequences of polar amplification elucidated by coordinated multimodel experiments" has now been seen by our reviewers, whose comments appear below. In light of their advice we are delighted to say that we are happy, in principle, to publish a suitably revised version in Communications Earth & Environment.

We therefore invite you to revise your paper one last time to address the remaining concerns of our reviewers. At the same time we ask that you edit your manuscript to comply with our format requirements and to maximise the accessibility and therefore the impact of your work.

EDITORIAL REQUESTS:

******Please take care to match our formatting and policy requirements. We will check revised manuscript and return manuscripts that do not comply. Such requests will lead to delays. ******

SUBMISSION INFORMATION:

OPEN ACCESS:

Communications Earth & Environment is a fully open access journal. Articles are made freely accessible on publication. For further information about article processing charges, open access funding, and advice and support from Nature Portfolio, please visit <https://www.nature.com/commsenv/open-access>

Link Redacted

Best regards,

ChenRui Diao, PhD

Associate Editor,
Communications Earth & Environment
Consulting Editor,
Communications Sustainability

REVIEWERS' COMMENTS:

Reviewer #1 (Remarks to the Author):

This paper summarizes much of the work of the Polar Amplification Model Intercomparison Project (PAMIP) to date. This project has led to major scientific advancements toward understanding how Arctic change broadly impacts the climate system through organization of multi-model, relatively large ensemble comparisons involving identical boundary states. Through this approach, the role of Arctic sea ice loss from natural variability has been isolated and key outcomes show tropospheric responses to sea ice loss including Northern Hemisphere-varying patterns of lower-tropospheric warming and wetting, equatorward shifts in the polar jet stream and North Atlantic storm track, and weaker cold extremes over North America.

The paper is well-written and graphics are nicely illustrated and presented. I have just a few minor recommendations centered around clarifying some of the terminology to make the paper more broadly readable. Comments below are by line (L) number of the submitted paper.

L59: What is meant by "asymmetries"? This is elaborated on in L330-347 but some brief context here would be helpful for clarity until the reader gets to concluding remarks where hemispheric comparisons are briefly made.

L158: Future Arctic sea ice loss is mentioned here and in areas of this section (L181, L191). The ice condition depends on the warming scenario, but can this reference be a little more clear with regards to changing seasonality, reduced areas/extents, and/or thinner sea ice packs?

L316: What is meant by storylines? A brief description is warranted to introduce this concept before it is expanded upon in the sentences that follow.

Figure 3: This is a very nice schematic, though would try to use consistent font colors in the captions to match colors of illustrated processes in the corresponding schematic. Most notably, the color match of the westerlies atmospheric profile (blue) and captions is different between panel 1 and 2.

Figure 4: It would be easier to read PAMIP-referenced studies if fonts were slightly enlarged and listed in black instead of gray.

Reviewer #2 (Remarks to the Author):

Review of "Causes and consequences of polar amplification elucidated by coordinated multimodel experiments" by JA Screen et al.

Overview: This is a Perspective piece summarising the knowledge gains resulting from the PAMIP exercise. PAMIP has been influential in the community, and a paper condensing its main outcomes is useful. I am therefore favorable to publication of this paper. I have a number of comments that may help further improve the manuscript.

Main comments:

1. Abstract: the title of the paper contains both "causes and consequences", but the abstract only summarises the consequences of polar amplification (PA) for midlatitudes. To properly reflect the title and content of the paper, the abstract should contain a couple of sentences summarising results on the causes of PA.

2. I.57-59: This sentence makes it sound like there are no outstanding questions regarding the qualitative functioning of PA, and the only remaining questions are quantitative (i.e., we know exactly how, but not exactly how much). I disagree with this

assessment. I think there is substantial remaining confusion in the community regarding, among other things: how feedbacks interact with each other; how lapse rate feedback should be interpreted; how to interpret the cancellation between dry and moist changes in atmospheric heat transport (does the cancellation mean AHT plays no role in PA?); how clouds change in the Arctic and how they affect PA. I recommend rephrasing in a more nuanced way that brings out these outstanding qualitative problems.

3. I.113-114: Actually the lapse rate feedback looks very small in Fig2a,b, comparable to or smaller than other feedbacks. I don't understand why it is particularly emphasised in this sentence. Suggest removing this sentence, or discussing all feedbacks of comparable importance and providing an overall assessment.

4. I.116-118: Well, yes, I agree that the increase in AHT due to midlatitude warming *by itself* causes only weak PA. But if the Arctic did not respond to remote SST warming (and warmed only because of sea ice retreat) then Arctic warming (and PA) would be much weaker. In other words, the roles of remote SST influence and local feedbacks are cumulative, and both give substantial contributions to PA. So I think this sentence gives a misleading summary of the situation. Please rephrase to give a more balanced summary, or provide arguments against the cumulative role of the remote and local processes.

5. I.139-140: This is another example of remote and local processes acting cumulatively to produce amplified changes in the Arctic, as discussed for temperature in point 4. Suggest mentioning this parallel (assuming you agree with the interpretation; if not, then explain why not).

6. I.195-198: Suggest adding a comment on how robust/credible/statistically significant these responses are.

7. I.293-295: Presumably this is the same response as found in the Smith et al (2022) paper discussed previously? I suggest making the link explicit to avoid confusion. Same comment applies to lines 298-299.

Minor comments:

I. 26: "because of energy budget changes and feedbacks triggered by diminishing sea-ice and snow cover": the phrase seems vague and a little redundant (since feedbacks will affect the energy budget). Try something more specific, e.g. "due to the combined effects of poleward energy transport and local feedbacks, particularly surface albedo feedback"

I.175-176: seems like this sentence is redundant, it just re-states what is said in the subsequent paragraph. Suggest removing this sentence.

I.188: Using the word "Lagrangian" to describe storm tracking is an abuse of language: "Lagrangian" means "following fluid parcel motion", and storms are not fluid parcels (their phase speed is different from that of the ambient wind, and air parcels can move in and out of storms). So please just say "storm tracking" if you mean storm tracking.

I.241: It could be useful to the readers to specify that the "eddy feedback" referred to here concerns the feedback of eddy momentum transport on the mean flow.

Reviewer #3 (Remarks to the Author):

Review on "Causes and consequences of polar amplification elucidated by coordinated multimodel experiments" by Screen et al., submitted to Communications, Earth & Environment

This perspective article summarizes the findings gained from the coordinated model experiment PAMIP (Polar Amplification Model Intercomparison Project) that was developed as part of CMIP (Coupled Model Intercomparison Project). Aims of PAMIP are to disentangle the influences of decreasing Arctic and Antarctic Sea Ice from the influence of increasing sea surface temperatures on Polar Amplification, the stronger-than-average winter near-surface temperature increase in the polar regions, and on polar-lower latitude linkages. One main result described in the present paper is "that Arctic amplification is caused primarily by sea-ice loss and resultant local changes in surface fluxes, while increased poleward energy transport can only produce weak amplification in the absence of sea-ice-related feedbacks". Another important result is that an increase in poleward latent heat transport assures that there is still some polar amplification even without sea ice loss. Both sea ice loss and increasing poleward latent heat transport lead to increasing Arctic precipitation but with regional differences. Also the second aim of getting a clear picture on the implications for mid-latitude climate is addressed in this perspective paper with clear graphics. Another important lesson from PAMIP is the role of the eddy feedback to intensify jet stream changes that may be underestimated in climate models.

The article is well structured and clearly written. Relevant work is cited. The article is timely as it suggests future directions for another PAMIP round for the next CMIP round. However, the title suggests an equal treatment of both hemispheres and is therefore triggering wrong expectations. It should either be adjusted as the article is focused on the Arctic and hardly gives any information on the Antarctic, except for in the future research direction section, or results on the Antarctic included. Given the timeliness and relevance of the perspective paper, I recommend publication in Communications, Earth & Environment, subject to adjustment of title or inclusion of Antarctic results and to very few minor comments below.

Minor comments:

Suggest swapping two rows of Fig. 2 since the focus is on sea ice changes (and the comparison is made to SST changes).
Future directions, Enhanced collaboration, second line: Exemplars -> Examples

** Visit Nature Portfolio's author and referees' website at www.nature.com/authors for information about policies, services and author benefits**

The reviewer comments are in blue, and our responses are in black. Revised passages are shown in green, with revisions in **bold and underlined**.

In addition to the changes suggested by the reviewers and editor, we have slightly modified the section ‘Reducing and exploiting uncertainty’, to include an important new paper published since our manuscript was submitted.

“Sigmond and Sun (2025) propose that the jet latitude response to future sea-ice loss can be constrained by the observed neck region winds, more than halving the model uncertainty⁶³. The neck wind constraint yields an estimated real world jet shift that closely matches the unconstrained multi-model mean response, implying that models do not systematically underestimate the jet shift⁶³, unlike the constraint based on observed eddy feedback^{36,37}. These contrasting findings underscore the need to test the robustness of such emergent constraints using independent ensembles.”

Reviewer #1 (Remarks to the Author):

This paper summarizes much of the work of the Polar Amplification Model Intercomparison Project (PAMIP) to date. This project has led to major scientific advancements toward understanding how Arctic change broadly impacts the climate system through organization of multi-model, relatively large ensemble comparisons involving identical boundary states. Through this approach, the role of Arctic sea ice loss from natural variability has been isolated and key outcomes show tropospheric responses to sea ice loss including Northern Hemisphere-varying patterns of lower-tropospheric warming and wetting, equatorward shifts in the polar jet stream and North Atlantic storm track, and weaker cold extremes over North America.

The paper is well-written and graphics are nicely illustrated and presented. I have just a few minor recommendations centered around clarifying some of the terminology to make the paper more broadly readable. Comments below are by line (L) number of the submitted paper.

We thank the reviewer for their time and constructive comments.

L59: What is meant by “asymmetries”? This is elaborated on in L330-347 but some brief context here would be helpful for clarity until the reader gets to concluding remarks where hemispheric comparisons are briefly made.

We meant differences in the character of polar amplification between hemispheres and between seasons. This has been rewritten, avoiding the word “asymmetries”.

“Thus, although the processes and feedbacks leading to polar amplification are reasonably well understood, **how they interact, their physical interpretation**, their relative contributions, and how they lead to **differences in the character of amplification between the hemispheres and seasons** are not precisely known”

L158: Future Arctic sea ice loss is mentioned here and in areas of this section (L181, L191). The ice condition depends on the warming scenario, but can this reference be

a little more clear with regards to changing seasonality, reduced areas/extents, and/or thinner sea ice packs?

We've added a sentence on this.

“More specifically, the PAMIP experiments impose changes in sea ice concentration (hence, areal coverage; Fig. 1) throughout the year and so, include shifts in the seasonal cycle, but not changes in sea ice thickness.”

L316: What is meant by storylines? A brief description is warranted to introduce this concept before it is expanded upon in the sentences that follow.

Storylines are a concept introduced by Shepherd et al (2018) to help communicate uncertainty in physical aspects of climate change. Storylines describe physically self-consistent plausible future pathways. In our case, they represent different possible combinations of global warming and sea-ice loss. We've added words to this effect in the revised manuscript.

“Storylines are a concept to help communicate uncertainty in climate change⁷⁸ and describe distinct but possible, physically self-consistent pathways. Here, the storylines describe plausible combinations of future global warming and sea-ice loss (Fig. 6), which are closely related to model uncertainty in the magnitude of Arctic amplification, expressed in terms of the magnitude of sea-ice loss per degree of global warming. The high sensitivity case describes a future with greater sea-ice loss relative to global warming, and the low sensitivity case describe a world with less sea-ice loss relative to global warming.”

Figure 3: This is a very nice schematic, though would try to use consistent font colors in the captions to match colors of illustrated processes in the corresponding schematic. Most notably, the color match of the westerlies atmospheric profile (blue) and captions is different between panel 1 and 2.

Well spotted! They were meant to be the same shade of blue. The colours now match.

Figure 4: It would be easier to read PAMIP-referenced studies if fonts were slightly enlarged and listed in black instead of gray.

We agree and have implemented this change.

Reviewer #2 (Remarks to the Author):

Overview: This is a Perspective piece summarising the knowledge gains resulting from the PAMIP exercise. PAMIP has been influential in the community, and a paper condensing its main outcomes is useful. I am therefore favorable to publication of this paper. I have a number of comments that may help further improve the manuscript.

We thank the reviewer for their time and constructive comments.

Main comments:

1. Abstract: the title of the paper contains both "causes and consequences", but the abstract only summarises the consequences of polar amplification (PA) for midlatitudes. To properly reflect the title and content of the paper, the abstract should contain a couple of sentences summarising results on the causes of PA.

Thanks for this suggestion. We've rewritten the open sentences of the abstract to better reflect the title and content.

"Human-induced warming is amplified in the Arctic, but its causes and consequences are not precisely known. Here, we review scientific advances facilitated by the Polar Amplification Model Intercomparison Project. Surface heat flux changes and feedbacks triggered by sea-ice loss are critical to explain the magnitude and seasonality of Arctic amplification."

2. I.57-59: This sentence makes it sound like there are no outstanding questions regarding the qualitative functioning of PA, and the only remaining questions are quantitative (i.e., we know exactly how, but not exactly how much). I disagree with this assessment. I think there is substantial remaining confusion in the community regarding, among other things: how feedbacks interact with each other; how lapse rate feedback should be interpreted; how to interpret the cancellation between dry and moist changes in atmospheric heat transport (does the cancellation mean AHT plays no role in PA?); how clouds change in the Arctic and how they affect PA. I recommend rephrasing in a more nuanced way that brings out these outstanding qualitative problems.

We agree and it was not our intention to imply otherwise. We have rephrased the sentence to emphasise that there is still a lot to learn. In the Future Directions section, we now advocate for the continued research into the interactions between feedbacks.

"Thus, although the processes and feedbacks leading to polar amplification are reasonably well understood, how they interact, their physical interpretation, their relative contributions, and how they lead to differences in the character of amplification between the hemispheres and seasons are not precisely known."

"Such modelling experiments offer opportunities to target remaining questions regarding the causes of polar amplification, such as the interpretation of the lapse rate feedback in polar regions."

3. I.113-114: Actually the lapse rate feedback looks very small in Fig2a,b, comparable to or smaller than other feedbacks. I don't understand why it is particularly emphasised in this sentence. Suggest removing this sentence, or discussing all feedbacks of comparable importance and providing an overall assessment.

Panels (a) and (b) in the original figure showed the response to SST change, whereas the text quoted refers to the response to SIC change, which was originally

shown in panels (c) and (d). When looking at the correct panels, the large contribution from the lapse rate feedback is clear. To avoid confusion, we have re-ordered the panels (as also recommended by reviewer 3) and now cite the specific panels in the text, rather than just the figure number. We have also reworded parts of this paragraph for clarity.

“This produces lower tropospheric Arctic warming and triggers positive lapse rate, **Planck**, and cloud feedbacks, leading to large Arctic amplification (**Fig. 2a,b**). Despite strong albedo feedback in summer, atmospheric warming is muted **in that season**. In contrast, in response to global SST warming absent sea-ice loss, enhanced atmospheric energy convergence into the Arctic is the dominant contributor to Arctic warming, although Arctic amplification is relatively small compared to that in response to sea-ice loss (**Fig. 2c,d**). **Here, the lapse rate feedback is of diminished importance consistent with SST-induced warming being larger aloft than at the surface. In all cases**, the water vapor feedback contributes to Arctic warming but opposes amplification due to larger tropical than Arctic moistening under SST-induced warming with fixed Arctic sea-ice. These results reinforce that **changes in surface fluxes and feedbacks triggered by sea-ice loss are critical to explain the magnitude and seasonality of Arctic amplification**, while increased poleward energy transport produces **weaker** amplification in the absence of sea-ice-related feedbacks.”

4. I.116-118: Well, yes, I agree that the increase in AHT due to midlatitude warming *by itself* causes only weak PA. But if the Arctic did not respond to remote SST warming (and warmed only because of sea ice retreat) then Arctic warming (and PA) would be much weaker. In other words, the roles of remote SST influence and local feedbacks are cumulative, and both give substantial contributions to PA. So I think this sentence gives a misleading summary of the situation. Please rephrase to give a more balanced summary, or provide arguments against the cumulative role of the remote and local processes.

While we believe that diagnostic decompositions are useful, a limitation is that they do not highlight interactions and cumulative effects. We have added a sentence to remind the reader of this limitation. In the subsequent paragraph, we add an example of a potential interaction.

“**We emphasise that Fig. 2 presents a diagnostic decomposition of the contributions to Arctic amplification and in reality, the effects of sea-ice loss and SST warming act in combination and interact.**”

“This process explains polar amplification in the absence of sea ice and related polar feedbacks^{4,24}, **and the greater amplification when local and remote drivers combine²².**”

5. I.139-140: This is another example of remote and local processes acting cumulatively to produce amplified changes in the Arctic, as discussed for temperature in point 4. Suggest mentioning this parallel (assuming you agree with the interpretation; if not, then explain why not).

Here we are simply making the distinction that SST-warming increases precipitation over a broad geographic region while sea-ice loss increases it preferentially in the

Arctic (see Figure 5). As stated at the beginning of the paragraph, both effects contribute to precipitation increases. We have edited the wording to make this clearer.

“Since SST warming alone leads to larger wetting at lower than higher latitudes, it is sea-ice loss that is critical for amplified wetting in the Arctic²⁹.”

6. I.195-198: Suggest adding a comment on how robust/credible/statistically significant these responses are.

Good suggestion. We now refer here to Fig. 4 that displays the confidence in each of these responses.

“Other weather system responses to Arctic sea-ice loss identified using the PAMIP experiments include an increase in winter Scandinavian blocking^{35,46}, a reduction in cut-off-lows over Southern Europe⁴⁷, enhanced tropical cyclone genesis over the eastern North Pacific⁴⁸, and shifts in the predominant locations of atmospheric rivers⁴⁹, **albeit with varying degrees of confidence (Fig. 4).”**

7. I.293-295: Presumably this is the same response as found in the Smith et al (2022) paper discussed previously? I suggest making the link explicit to avoid confusion. Same comment applies to lines 298-299.

These results are related, but not the same. Smith et al. (2022) presented a zonal-mean perspective, whereas here we are talking about regional responses. We've clarified this.

“Using the PAMIP experiments, Yu et al. (2024) demonstrated a significant **regional weakening of the westerlies over the high-latitude North Atlantic and strengthening over the midlatitude North Atlantic in response to Arctic sea-ice loss²⁸, **consistent with the zonal-mean response**³⁶, while ocean warming led to a broadly opposite response²⁸ (Fig. 5). This regional ‘tug-of-war’ in the Atlantic is also seen for storm track activity^{38,50}, but not in the Pacific^{28,38,41}. Hay et al (2025) showed that more equatorward average location of the Pacific jet and the geography of the North Pacific results in circulation responses to sea-ice loss and ocean warming that reinforce rather than oppose each other³⁸, **in contrast to the zonally-averaged response**³⁶.”**

Minor comments:

I. 26: "because of energy budget changes and feedbacks triggered by diminishing sea-ice and snow cover": the phrase seems vague and a little redundant (since feedbacks will affect the energy budget). Try something more specific, e.g. "due to the combined effects of poleward energy transport and local feedbacks, particularly surface albedo feedback"

We've rewritten this part of the abstract. See earlier response.

I.175-176: seems like this sentence is redundant, it just re-states what is said in the subsequent paragraph. Suggest removing this sentence.

Done.

I.188: Using the word "Lagrangian" to describe storm tracking is an abuse of language: "Lagrangian" means "following fluid parcel motion", and storms are not fluid parcels (their phase speed is different from that of the ambient wind, and air parcels can move in and out of storms). So please just say "storm tracking" if you mean storm tracking.

Storm tracking is often described as "Lagrangian", to differentiate it from grid-point (Eulerian) metrics. However, we have simplified this sentence, avoiding the use of the wording "Lagrangian" and "Eulerian".

I.241: It could be useful to the readers to specify that the "eddy feedback" referred to here concerns the feedback of eddy momentum transport on the mean flow.

This is a good suggestion, thanks, and has been implemented.

Reviewer #3 (Remarks to the Author):

This perspective article summarizes the findings gained from the coordinated model experiment PAMIP (Polar Amplification Model Intercomparison Project) that was developed as part of CMIP (Coupled Model Intercomparison Project). Aims of PAMIP are to disentangle the influences of decreasing Arctic and Antarctic Sea Ice from the influence of increasing sea surface temperatures on Polar Amplification, the stronger-than-average winter near-surface temperature increase in the polar regions, and on polar-lower latitude linkages. One main result described in the present paper is "that Arctic amplification is caused primarily by sea-ice loss and resultant local changes in surface fluxes, while increased poleward energy transport can only produce weak amplification in the absence of sea-ice-related feedbacks". Another important result is that an increase in poleward latent heat transport assures that there is still some polar amplification even without sea ice loss. Both sea ice loss and increasing poleward latent heat transport lead to increasing Arctic precipitation but with regional differences. Also the second aim of getting a clear picture on the implications for mid-latitude climate is addressed in this perspective paper with clear graphics. Another important lesson from PAMIP is the role of the eddy feedback to intensify jet stream changes that may be underestimated in climate models.

The article is well structured and clearly written. Relevant work is cited. The article is timely as it suggests future directions for another PAMIP round for the next CMIP round. However, the title suggests an equal treatment of both hemispheres and is therefore triggering wrong expectations. It should either be adjusted as the article is focused on the Arctic and hardly gives any information on the Antarctic, except for in the future research direction section, or results on the Antarctic included.

Given the timeliness and relevance of the perspective paper, I recommend publication in *Communications, Earth & Environment*, subject to adjustment of title or

inclusion of Antarctic results and to very few minor comments below.

We thank the reviewer for their time and constructive comments.

Minor comments:

Suggest swapping two rows of Fig. 2 since the focus is on sea ice changes (and the comparison is made to SST changes).

Good idea. We've implemented this suggestion.

Future directions, Enhanced collaboration, second line: Exemplars -> Examples

Changed.